# Caregiving motivations and experiences among family caregivers of patients living with advanced breast cancer in Ghana

Grace Kusi[1,2], Adwoa Bemah Boamah Mensah[1]*, Kofi Boamah Mensah[3], Veronica Millicent Dzomeku[1], Felix Apiribu[1], Precious Adade Duodu[1]

1 Department of Nursing, Faculty of Allied Health Sciences, College of Health Sciences, Kwame Nkrumah University of Science and Technology, Kumasi, Ghana, 2 Department of Obstetrics and Gynecology, Komfo Anokye Teaching Hospital, Kumasi, Ghana, 3 Oncology Directorate, Komfo Anokye Teaching Hospital, Kumasi, Ghana

* bbemahc2000@gmail.com

## Abstract

### Introduction

Significant number of women present with advanced-stage breast cancer in Ghana. These women usually depend on family caregivers for their multi-dimensional needs. Yet, there are gaps in research about what motivates family caregivers to assume the caring role and their experiences with caregiving within the Ghanaian context.

### Aim

To explore and describe the caregiving motivations and experiences among family caregivers of patients living with advanced breast cancer.

### Methods

In-depth, semi-structured qualitative interviews were conducted with 15 family caregivers who were providing unpaid care for women living with advanced breast cancer. Colaizzi's thematic analysis was used to analyze the data.

### Results

Family relationship normally prescribed the caregiving role among family caregivers. Due to the lack of home-based palliative services in Ghana, findings suggest that family caregivers are the main managers of advanced breast cancer-related symptoms in the home. These findings are discussed under three major themes: (i) motivation for assuming the caregiving role; (ii) meeting self-care and psychosocial needs of the patient; and (iii) symptom management and monitoring.

### Conclusion

Socio-cultural values influence the role of family caregivers in Ghana. This presents opportunities for health professionals and relevant stakeholders to develop a culturally-

**Data Availability Statement:** All relevant datasets on which conclusions of this manuscript were made are available as a Supporting Information file

and in the Figshare repository (https://figshare.
com/articles/Caregiving_motivations_and_
experiences_among_family_caregivers_of_
patients_living_with_advanced_breast_cancer_in_
Ghana/11857977).

**Funding:** The authors received no specific funding
for this work.

**Competing interests:** The authors have declared
that no competing interests exist.

appropriate intervention to support informal caregivers in their home-based care for women
living with advanced breast cancer in Ghana.

## Introduction

In Ghana, breast cancer is currently the leading cause of cancer deaths among women [1]. It
has been estimated that about 4,700 women were diagnosed with breast cancer and about
1,900 mortalities were recorded in 2018 [1]. The most disturbing pattern is that the majority of
Ghanaian women (50–80%) report with advanced disease (stage III and IV) [2–5]. These
women often experience devastating symptoms such as pain, lymphedema, and breast wound
[6]. Given the stark burden of symptoms and the sufferings faced by Ghanaian women living
with this disease, they are usually in need of various anti-cancer treatment modalities including
palliative care services [6]. Palliative care is an approach that improves the quality of life of
patients and families who are facing the challenges associated with life-threatening illness such
as advanced breast cancer [7]. Although palliative care does not focus on cure, the goal of palli-
ative care is achieved through the prevention and relief of suffering, by means of early identifi-
cation, assessment and treatment of pain and other physical, psychosocial and spiritual
distress [7]. Palliative care is an essential component of comprehensive healthcare services for
patients and families with non-communicable disease such as breast cancer [7] and employs
various models of delivery including the home-based model [8]. Home-based palliative care is
a globally recognized model of health service provision which was introduced by the World
Health Organization (WHO) to improve the quality of life, care, and death as well as promote
patient and family's satisfaction [9]. The model further decreases burnout in staffing and mor-
tality in hospitals; reduces the cost, accepts end of life as live days; neither accelerates death nor
prolongs life; considers all dimensions of human; helps the patient to be active until the time
of death; helps the family to cope with the disease and loss of the patient [9].

However, in the Ghanaian context, the home-based model of care is just emerging and the
most commonly practiced model is the out-patient palliative care model since the treatment
trend for breast cancer as well as palliative care is gradually transitioning to out-based settings
[10]. Further, collaboration among interdisciplinary team members such as physicians, oncol-
ogists, wound specialists, pharmacists, psychologists, social workers, nurses, and other relevant
disciplines which is an important feature of palliative-care is lacking in the Ghanaian context
[11]. Consequently, the caregiving responsibilities fall on family members and other significant
relations in the home setting. These caregivers assume the caregiving roles to support and
address the multi-dimensional needs of women living with advanced breast cancer using the
critical resources available in the home setting [6,12].

Previous studies have reported that in the home-setting, caregivers are often responsible for
providing treatment decision-making, symptom management, physical as well as emotional
support [13,14]. Traditionally, caregiving has also been described in terms of the functional
status of the patient [15]. For instance, caring for women with progressive breast disease
demands great involvement from family caregivers which requires them to perform more
medical roles and also adopt skills that enable them to respond better to the physical, psycho-
social and emotional needs of the patients [16]. According to Kristanti et al [17], caregivers'
commitment to performing caregiving duties for individuals with terminal illnesses such as
advanced breast cancer depends on their motive for accepting the caregiving role.

Prior research has highlighted that factors such as social norms and cultural values may
influence caregivers' choices to accept the caregiving role [17–21]. For instance, a study by Ng
et al among caregivers in Singapore found that familial commitments were reasons why

caregivers assumed the caregiving role for breast cancer patients [21]. However, these developed countries differ in their experiences of caregiving compared to developing countries such as Ghana where supportive structures and systems seem not available for the family caregiver of the advanced breast cancer patient. Yet, there is a significant gap in the literature about what influences the decisions of family caregivers to assume such roles in such a resource-limited country, Ghana [18]. In this study, a family caregiver refers to a relative or friend who provides unpaid home care management to the advanced breast cancer patient.

Given the high symptom burden and the challenging needs of Ghanaian women diagnosed with advanced breast cancer [6], it is vital to explore what primarily motivates individuals to assume such challenging roles for women with this disease despite the progressive nature of the disease. Given the lack of evidence on the phenomenon within the Ghanaian context, this study aimed to explore the caregiving motivations and experiences of family caregivers of patients living with advanced breast cancer. Understanding these experiences is imperative, as it will potentially guide the development of a collaborative and contextualized home-based care model to support family caregivers in their caregiving roles accordingly.

### Aim

To explore and describe the caregiving motivations and experiences among family caregivers of patients living with advanced breast cancer.

## Materials and methods

### Study design and setting

An exploratory descriptive phenomenological approach was used to explore and describe the motivations and experiences of participants [22]. Phenomenology, as used in nursing research, is concerned with understanding the lived experiences of individuals regarding a particular phenomenon [23]. The study was conducted at the Komfo Anokye Teaching Hospital (KATH). The hospital is the second-largest hospital in Ghana and has a national center for nuclear medicine and oncology with palliative care services for patients diagnosed with breast cancer in the Ashanti Region and beyond. Specifically, the oncology unit of KATH, the recruitment outlet, was chosen for the study because it serves as the main referral center for the northern sector of Ghana and beyond. The female cancer most commonly diagnosed in the region and seen at KATH is breast cancer, with almost 80% of women presenting with advanced breast cancer [5,24]. These features made the hospital suitable for this study.

### Population, sampling technique, and sample size

Caregivers of women diagnosed with advanced breast cancer in the Ashanti Region, Ghana, were the target population for this study. To be recruited, participants satisfied the following inclusion criteria: (i) ≥ 18 years and above; (ii) family caregiver of a patient diagnosed with advanced breast cancer (Stage III & IV) for at least three months; (iii) providing unpaid care services to the patient; and lastly (iv) could speak either English or Twi. Caregivers of patients diagnosed with early-stage breast cancer were excluded from this study.

A purposive sampling technique was used. This helped the researchers to enroll participants who had experienced the phenomenon under study and met the inclusion criteria into the study [25–27]. A total of 24 potential participants were approached, but 19 voluntarily expressed interest in the study. However, 15 participants were included in the study based on data saturation. Data saturation is the point where all themes were fully developed and no new

insights evolved from the data [28]. All the participants gave consent by signing (10) or thumb-printing (5).

## Recruitment procedure

The head of the oncology unit gave formal permission to the authors for the study. This permission was granted after giving an introductory letter that specified the purpose of the study and a copy of the ethical approval to the head of the unit. There were pre-data collection interactions with the staff to discuss the study. Three nurses ("recruitment links") at the oncology unit were identified to aid in recruiting participants. Subsequently, the nurses were met to discuss the recruitment process, detailing the selection criteria. They were provided with the recruitment tool bearing the researchers' contact details. The nurses presented potential participants with study information sheets. For prospective participants who were unable to read and understand, the "recruitment nurses" read and translated the content of the information sheets to them.

The first author was contacted when potential participants were identified and their phone numbers were provided by the recruitment nurses. Potential participants were then contacted confidentially via phone to assess their suitability for the study by the first author and a final screening was done to determine their eligibility for the study based on the inclusion criteria. The recruitment process came to an end after the 15th interview following data saturation [28].

## Data collection tool and procedure

A semi-structured interview guide (S1 File) was used as the data collection tool. This was developed by the authors based on existing literature and the study objectives. The guiding questions were reviewed by oncology and palliative care specialists as well as the second author, a qualitative researcher with clinical experience in oncology. The first author consulted an expert who spoke and wrote Twi (local language) to translate (back-back translation) the interview guide.

Before the actual study, the guiding questions were piloted at a private oncology center in Kumasi with two family caregivers who were not involved in the final study. This center has similar geographical, socio-economic and cultural features with KATH. The outcome of the pilot study was used to address practical concerns of the guiding questions accordingly. However, the results of these pilot interviews were not included in the main results of this study.

Pre-interview demographic information was collected before the main interview. A clinical psychologist was consulted to offer counseling services to participants who needed emotional support after sharing their experience. However, none of the participants used this service. Eleven (11) participants had the main interview conducted in their homes while four (4) participants preferred to be interviewed at the hospital. Non-participants were not allowed to the venue for the interviews. Further probes and paraphrasing were used to understand the participants' caregiving motivations and experiences. The mean duration of the interviews was 80 minutes (ranging from 45 to 115 minutes). Interviews were audio-recorded with participants' consent. Interviews were conducted in English (n = 1) and Twi (n = 14). No repeat interviews were conducted.

The researchers listened and transcribed the audiotapes concurrently to help explore the topic in successive interviews. The first author kept field notes for each interview. These field notes recorded the participants' non-verbal cues, concerns, and researchers' reflections. After each interview, member checks were done with each participant by summarizing the responses to ensure an accurate representation of the participants' views. This maximized the credibility of the study. Also, transcripts were returned to four participants to read through to ensure it reflected their views.

## Data analysis and management

Data was analyzed using Colaizzi's rigorous and robust method of analysis to find, understand, describe, and depict [29] the motivations and caregiving experiences of family caregivers of women diagnosed with advanced breast cancer. The seven step processes used for the data analysis as described by Colaizzi are detailed below.

*Step 1*. All the interviews were transcribed verbatim by the first author in their respective languages; Twi (14) and English (1). Two independent translators fluent in both the Twi and English languages translated the 14 anonymized "Twi" transcripts using the process of back-back translation while maintaining confidentiality. The transcripts for all the 15 participants were independently read repeatedly by the first author (GK) and the second author (ABBM) to make sense of the caregiving motivations and experiences among the family caregivers of patients living with advanced breast cancer.

*Step 2*. To generate data that directly pertain to caregiving motivations and experiences among family caregivers of patients living with advanced breast cancer, significant recurring statements or phrases were extracted from participants' transcripts as codes.

*Step 3*. Formulated Meanings were then created from the significant statements or phrases to describe and illuminate the meanings hidden in the various contexts of the phenomenon under study. Subsequently, themes were generated based on the multiple statements that conveyed similar meanings.

*Step 4*. Step 1 to step 3 were then repeated for all the 15 transcripts to identify the experiences that were common to all participants. All the formulated meanings were then categorized into clusters of themes based on their similarities and relationships.

*Step 5*. A database was created to compile all the exhaustive descriptions that were generated in step 1 to step 4.

*Step 6*. The results were then summarized and incorporated into a rich and exhaustive description of the caregiving motivations and experiences of participants. The field notes were also used to corroborate the themes developed.

*Step 7*. To ensure the credibility of the findings, transcripts were returned to four (4) participants to confirm whether their realities were truly captured. However, no new or pertinent data were obtained.

The researchers assigned participants with numbers to protect their anonymity. For confidentiality, the anonymized transcripts were saved on a password-protected computer and copied on a pen drive to prevent data loss.

## Rigor and reflexivity

We ensured the trustworthiness of the study by applying the principles of credibility, dependability, confirmability, transferability, and authenticity [30,31]. Member checks at the end of each interview were used to fully understand and correctly present the respondents' stories and to ensure credibility. Credibility was also ensured through the independent analysis and review by the authors. The authors provided an exhaustive study methodology description (recruitment procedure, data collection, data analysis, etc.) to achieve study replicability or dependability [32]. For transferability, detailed descriptions of the research context, participants' backgrounds, and study results were explicitly provided to enable other researchers to transfer the findings to their context and setting.

According to Fontana, reflexivity is a pillar of 'critical' qualitative research [33]. Also, it is described as a quality control measure to reduce the subjectivity of the researcher [34]. To ensure confirmability, field notes detailing verbal and non-verbal cues recorded during the interview were used to corroborate the transcripts[30, 31]. In addition, the data collection

process and preliminary analysis were done simultaneously, as well as the verification of all back-to-back transcriptions from Twi to English and vice-versa which were done by two translators.

GK (first author) conducted all the interviews due to her ability to speak and write in both the 'Twi' and English languages. The interviewer is a female nurse who works at the Department of Obstetrics and Gynecology of KATH. Nevertheless, she is not related to any of the participants and had no direct influence on the study setting and results. To ensure authenticity, data were analyzed inductively [35] to describe the caregiving motivation and experiences as narrated by the participants. Also, field noted were taken and used to give context to the data.

## Ethical considerations

The study obtained ethical approvals from the Ethics Review Board (Kwame Nkrumah University of Science and Technology; CHRPE/AP/571/18) and the Research and Development Unit, KATH also approved and registered the study (RD/CR18/235). The purpose of the study, its benefits and risks were explained to the participants during the recruitment stage for them to willingly decide to participate in the study. Although there were no direct or indirect benefits for study participants, they were educated that their participation will contribute to improved caregiving to women diagnosed with advanced breast cancer. Participants consented to participate in the study by writing. Participants were assured of their anonymity in future publications and reports. All data will be destroyed after 5 years of this study.

## Results

### Participants

A sample of 15 family caregivers was recruited into the study. Participants were aged between 25 to 73 years. Participants consisted of seven (7) men and eight (8) women. Eleven (11) of the participants cared for patients living with stage III breast cancer whereas four (4) of the participants cared for women with stage IV breast cancer. The detailed characteristics of the participants are shown in S1 Table.

### Main findings

In Ghana, hospital-based palliative care for cancer patients is regrettably inadequate, thereby creating an avenue for family members and significant others to assume the caregiving role for advanced breast cancer women. The current study affords unique insights into caregiving motivations and experiences of family caregivers of women living with advanced breast cancer. Three major themes that emerged from the study were motivation for assuming the caregiving role, meeting self-care and psychosocial needs of patients, and symptom management and monitoring as summarized in Table 1 below:

### Motivation for assuming the caregiving role

This theme describes the motive that guided participants' decision to become the primary caregivers of women living with advanced breast cancer. The underlying sub-themes under this theme were: (1) caregiving as a family and socio-cultural obligation; and (2) caregiving as a means of reciprocity.

**Caregiving as a family and socio-cultural obligation.** Participants in this study cited that the family's obligation to care for its ill individuals was the main reason that underpinned their decision to accept the caregiving role. The family, a recognized institution in Ghana [36], is usually seen as the main source of support for individuals in the Ghanaian socio-cultural

**Table 1. Themes and sub-themes generated.**

| THEMES | SUB-THEMES |
|---|---|
| Motivation for assuming the caregiving role | • Caregiving as a family and socio-cultural obligation<br>• Caregiving as a sign of reciprocity |
| Meeting self-care and psychosocial needs of patients | • Assisting with activities of daily living<br>• Spiritual support<br>• Emotional support<br>• Financial support |
| Symptom management and monitoring | • Home-based wound care<br>• Management of breast cancer-related lymphedema<br>• Drug administration and pain management<br>• Continuous evaluation of symptoms and patient advocacy |

context during sickness and birth [37]. Thus, this cultural norm informed this study's participants to regard caregiving to their relatives as a principal socio-cultural obligation. This is apparent in the following quotes:

> *"She is my mother, my family. . . it is actually my socio-cultural responsibility to take care of her. . . That is the reason I am the one taking care of her" (Participant 14, Son).*

> *"As our culture demands, I have to take care of her. It is just my duty as her (patient) family member" (Participant 7, Sister).*

The cultural orientation of Ghana towards the female gender also assigns the caring role for homes, children and sick relatives specifically to mothers and older daughters [38–40]. Hence, some participants who were women also voiced out that it was culturally expected of them as mothers and older daughters to take care of ill family members such as women with advanced breast cancer as illustrated in the quote below:

> *"Hmmm..! As the eldest daughter, it is my cultural duty to take care of her. I have no other option than to take care of her" (Participant 12, daughter).*

> *"I am a woman and her mother and it is my cultural duty to take care of her" (Participant 11, Mother).*

In the absence of female family members who are the predominant caregivers, some men also assumed the caregiving roles for women diagnosed with advanced breast cancer. In the Ghanaian cultural context, spouses offer vital support to their partners in all life situations including sickness, and this includes the essential role played by men in the medical decision-making of their wives [41]. In this study, participants who were spouses of the patients also cited marital obligation as a vital societal expectation which prompted them to be caregivers. As one spouse narrated:

> *"Oh! She is my wife. . . if I don't take care for her, who will? It is just my social obligation as a husband to take care of her" (Participant 10, husband).*

A brother of one of the patients also said:

> *"When she was diagnosed of the disease (breast cancer) she had no one to take care of her, hence, as a brother, it is culturally expected that I take care of her. There are no women around, hence I have to do it" (Participant 3, brother).*

**Caregiving as a means of reciprocity.**   Some participants were primarily motivated by reciprocity to assume the caregiving role. Participants cited that caregiving was a means of repayment of a good deed the participant had initially obtained from the patient. This is shown in the following narratives:

*"It is unusual for someone who is not your family member to take care of you when you are sick, but she (patient) did it passionately when I was admitted at the hospital. So now that she is sick, it is my turn to repay her for what she did for me"* (Participant 9, Friend).

*"She used to support me by lending me money when she was working. So now that she is sick, I have to also support her till she recovers from this illness. I am just paying her back for all the support she has been giving me"* (Participant 15, Friend).

## Meeting self-care and psychosocial needs of patients

This theme explores the participants' experiences in supporting the patients to meet their daily needs. Further, this theme describes the support that is provided by participants in meeting the spiritual and psychosocial needs of patients living with advanced breast cancer. Four sub-themes emerged under this theme: (i) assisting with activities of daily living; (ii) spiritual support; (iii) emotional support, and (iv) financial support.

**Assisting with activities of daily living.**   Participants described that patients with advanced breast cancer were unable to perform physical activities such as bathing, cooking, and grooming due to symptoms such as a wound, weakness, lymphedema, and treatment side effects. Hence, as part of their caregiving role, participants assisted patients with their daily activity needs.

*"I fetch and boil water for her every day. I also groom her every day because she becomes very weak whenever she goes for therapy (chemotherapy)"* (Participant 14, Son).

*"She cannot wash her dirty cloths because her hand is always swollen and heavy (lymph-edema). So, I am the one who does all her laundry"* (Participant 3, brother).

**Spiritual support.**   This sub-theme addresses the support that was offered by participants to patients to meet their spiritual needs. Generally, spirituality and religion are said to play key roles among cancer patients [42], and this was reported in a previous study in Ghana [6]. Participants in this study described that they aided patients to address their spiritual concerns through prayer, sharing the word of God with them and encouraging their faith in God. This is evident in the expressions below:

*"I pray for her and share healing messages in the Bible with her. This has really helped her to have some inner peace now"* (Participant 11, Mother).

*"Now she (patient) does not cry anymore because I always encourage her that God is on the throne and that He will heal her. I pray and share God's words with her. These have really increased her faith in God"* (Participant 4, Husband).

Other participants also cited that they ensured that patients had access to their spiritual leaders and benefited from spiritual services such as the Holy Communion in the home. This

is because spiritual leaders are key stakeholders during illnesses including breast cancer diagnosis in Ghana [3]. Two participants had this to say:

> *"I always remind the church leaders every month about the Holy Communion. I make sure that they always bring her bread and wine in the house" (Participant 9, Friend).*

> *"Sometimes, I arrange pastoral visits for her. The Osofo (Reverend) comes in to pray with her (Participant 13, Brother).*

**Emotional support.**   Participants described that advanced breast cancer disease affected patients emotionally. Emotional support was therefore perceived as being critical in aiding patients cope with advanced breast cancer disease. Hence, in this regard, participants described that they provided words of encouragement as shown in the following narrations:

> *"She was always thinking about the disease, but I encourage her with cheerful and hopeful messages to relieve her emotional distress" (Participant 6, Sister).*

> *"I always make sure that I communicate with her and encourage her to forget about the breast cancer so that she can be happy all the time" (Participant 15, Friend).*

Participants consoled the patients any time they cried or became worried about their condition by keeping a positive attitude. This is corroborated as below:

> *"Although I worry a lot about her condition every day, I try to be cheerful when I am with her. I get time to listen to her and console her when she is lonely" (Participant 3, Brother).*

**Financial support.**   Participants also narrated that they provided financial support to cover for treatment cost since there is limited coverage of breast cancer treatment by the National Health Insurance Scheme (NHIS). The Ghana NHIS provides partial coverage for the treatment cost of breast cancer [43], therefore, advanced breast cancer patients and their caregivers bear most of the treatment costs. Since most of the participants had no income-generating jobs due to the debilitating progression of the disease, their caregivers (participants) mostly catered for the patients' medical expenses.

> *"At the hospital, all the little money on me has been spent on her treatment because the NHIS covers only the folder and some of the infusions but all the other medications, I have to buy them" (Participant 1, Husband).*

> *"The National Health Insurance Scheme (NHIS) does not cover all the cost of the treatment. So, the little I get, I spend it all on my mother's drugs and living expenses" (Participant 2, Son).*

Participants also cited that they also provided financial assistance for other non-treatment costs such as transportation for access to medical care.

> *"I provide money for everything including transportation to the hospital. Where we live, if we take a taxi to the hospital, they charge GHC 120 (≈23 USD) for our round trip" (Participant 10, Husband).*

*"Aside money for her drugs, I always buy food supplements (Forever Living) to meet her nutritional needs. Although expensive, I manage to buy it for her" (Participant 15, Friend).*

## Symptom management and monitoring

This theme describes the experiences of participants in the management of symptoms of advanced breast cancer and its related treatment side effects. Ghana, a country that lacks resources, is also burdened with inadequate cancer care centers [44], inappropriate integrative palliative care services [6], and gross geographical barriers to cancer treatment [45]. Limited dressing facilities, as well as a lack of wound care specialists exists in the Ghanaian health care system [46]. Thus, advanced breast cancer patients depended on their caregivers for the management of their symptoms.

In this study, the sub-themes generated under this theme are (i) home-based wound care; (ii) drug administration and pain management; (iii) management of breast cancer-related lymphedema; and (iv) continuous evaluation of symptoms and patient advocacy.

**Home-based wound care.** Participants cited that they were the sole agents in the management of malignant wounds in the home setting. Participants described that they resorted to the services of traditional healers, socio-culturally acceptable in the Ghanaian health care system [47], in managing malignant wound symptoms such as odor, discharge, and bleeding in the home. This is illustrated in the narratives below:

*"We went to an herbalist who gave us a mixture of charcoal and clay to apply on the wound to help with the bleeding. It really helped with the bleeding and helped in removing the sloughs. So I use it for the dressing at home" (Participant 11, Mother).*

*"I sent her to an herbal center at Asokwa. The herbal doctor usually grinds some green leaves for us to apply on the wound at home and it has really reduced the mal-odor" (Participant 9, Friend).*

**Drug administration and pain management.** Participants cited that in the home setting, they used prescribed analgesics to manage patients' cancer-related pains. Nonetheless, participants narrated that these prescribed analgesics were not adequately alleviating the breast cancer-related pains. Hence, participants modified the prescribed dosages of these analgesics to aid in relieving breast cancer-related pains.

*"Per the doctor's instruction, she was supposed to take 100 mg tramadol 2 times daily and 1 gram paracetamol three times daily. But the pain was still unbearable so I had to give her more than the prescribed medicine. Now, I give her 200mg tramadol to relieve her pain and also help her to sleep" (Participant 12, Daughter).*

*"One tablespoon (10mg) Morphine 3 times daily is what they (doctors) prescribed for her. But, normally, she complains of pain in the breast. The breast is ulcerated so she really feels the pain. So I administer two tablespoons (20mg) to her in the house anytime she complains of pain. I don't stick to the prescribed dosage because it just couldn't relieve her pain" (Participant 4, Husband).*

Others also used non-medical means to control the patients' pains:

*"Where we were staying, we had no fan. So whenever she cried and complained of pain, I fan her to reduce the pain and further promote her comfort" (Participant 10, Husband).*

*"Sometimes when she complains of pain, I intentionally play her favorite program on TV for her to watch and it really helps to take her mind off the pain"* (Participant 5, Sister).

Further, participants described some adverse effects of prescribed opioids such as addiction, sedation, and hallucinations. Hence, to avoid the occurrence of these side effects, they administered the prescribed dosages only in situations when patients suffered from extreme pain.

*"Sometimes, she (patient) can ask for the whole bottle of morphine because of the unbearable pain. But I always say no to her request. The doctor says you can experience side effects such as constipation and addiction when you take too much of the morphine"* (participant 8, daughter).

*Due to this, I do not administer the drug daily per the prescription . . . I administer the drug to her when the pain is too much (Participant 11, Mother).*

**Management of breast cancer-related lymphedema.** The majority of the participants made it known that they were also in charge of managing lymphedema and its related symptoms. Participants used different strategies to reduce the lymphedema as described below:

*"Her arm is always heavy so I usually apply bandages and put the arm in a sling for her. Now the weight has started reducing gradually"* (Participant 12, Daughter).

*"When her arm becomes swollen, I make sure I elevate the affected arm by putting the arm on pillows. This has helped in decreasing the swollen arm"* (Participant 4, Husband).

Participants also cited that they used various approaches such as traditional approaches in managing breast cancer-related lymphedema in the home setting.

*"Her arm was always swollen and I sent her to an herbalist who put some herbs in a horn and blew it on the swollen arm"* (Participant 10, Husband).

*"I usually mix charcoal with clay and apply it on her (patient) swollen hand and by the next day, the hand will be better. So, that is what I have been using for the hand"* (Participant 11, Mother).

**Continuous evaluation of symptoms and patient advocacy.** This sub-theme describes participants' experiences in monitoring treatment-related side effects such as vomiting and subsequently communicating with health professionals about these treatments induced side effects. This is evident in the following quotes:

*"After the first cycle of chemotherapy, they told us she will vomit and also have diarrhea. But the vomiting was too much for three consecutive days. So, I called the doctor and told him about it and he said it is the side effect of the chemotherapy. But I told the doctor that the vomiting was too much. So, he told us to send her to the nearest hospital"* (Participant 6, Sister).

*"I don't know whether it was because of the drugs she was taking. She was always complaining "my stomach, my stomach". When she complains of the stomach pains, she usually feels like vomiting so I called the doctor and told him about what was happening"* (Participant 11, Mother).

## Discussion

The motivations identified for providing care to the women with advanced breast cancer included family and socio-cultural obligation and reciprocity. Family and socio-cultural obligation reflect the cultural norm that is embedded in the Ghanaian society which views the family as the key provider of support during terminal illnesses such as breast cancer [37].

Participants described that caregiving was entrenched in the Ghanaian cultural values; hence, the decision to care was viewed as an automatic obligation. This finding supports earlier evidence that describes caregiving as a communal family activity that is shaped by cultural norms [19]. Further, from our study, there were gender prescriptions in which females, specifically mothers and older daughters, were supposed to assume the caring role. This is because, in the Ghanaian socio-cultural context, females are usually seen as "domestic beings" [40]. Additionally, our findings indicated that there were marital obligations in which male spouses were the main primary caregivers. This evidence was further confirmed in Yeung et al's [19] quantitative study of male spouses who cited that marital obligation may influence the decision to accept the role of caregiving for women living with advanced breast cancer. Also, our finding that caregiving was based on reciprocity supports previous research which showed that caregiving can be viewed as a form of repaying of good deeds formerly obtained from patients [48].

Study findings showed that, in addition to a motivation for caregiving, caregivers also assisted with the daily needs of patients such as bathing, grooming, and cooking. The progressive nature of advanced breast cancer and its treatment resulted in declining functional status which resulted in patients being unable to perform daily living activities such as grooming, groceries, and bathing. Our findings correspond with earlier research works which posit that caregivers of women diagnosed with advanced breast cancer provided support with activities of daily living [14, 49].

Further, findings in the study highlighted the experiences of caregivers in symptom management such as the management of pain, lymphedema, wound, and evaluation of symptoms. In the management of wounds, participants narrated that they used herbal products, charcoal, and coal to manage wound symptoms such as odor, bleeding, and discharge. This finding could be explained by the fact that there is a lack of home-based palliative care services and wound care specialists in the Ghanaian health system [46], consequently making caregivers the main managers of the wound in the home setting. This finding, thus, has implications for health professionals especially nurses to educate caregivers on how to manage malignant breast wounds in the home to prevent the potential occurrence of breast wound infections.

In addition, in the management of pain, participants also cited that they normally altered the prescribed pain medications to improve the comfort of patients. This finding indicates that caregivers should be given support that can aid them in addressing their breast cancer pain management concerns. Further, findings in the current study showed various dimensions in which caregivers provided support to patients. Caregivers were the main providers of emotional support by offering patients with words of encouragement. Previous authors confirmed that emotional support from the family is important for women with advanced breast cancer due to the emotional and psychological impact on women [6]. Providing spiritual support through prayers and organizing for pastoral visits coupled with the Holy Communion for patients were other dimensions of support that caregivers offered for patients. As reported in prior studies, spiritual practices such as prayers are profoundly rooted in the Ghanaian context, and this further move individuals to trust in God in difficult times including periods of terminal illnesses [6, 50]. Hence, this might have urged caregivers to offer spiritual support to patients. This finding has implications for the role of spiritual bodies in the provision of home-based support to families affected by advanced breast cancer.

In the Ghanaian health structure, it is recognized that breast cancer and other women cancers are fully covered under the National Health Insurance Scheme (NHIS). However, it is perceived that the NHIS only provides partial subsidizations for breast cancer treatment, making patients and their family members the main financial providers of treatment costs [43]. This perception was confirmed in this study, as caregivers reported that they financially supported the patients by providing out of pocket money for treatment costs due to limited availability of NHIS funded therapies and other related non-medical costs such as providing transportation for accessing treatment. Consequently, there is a need for the government to address the deficiencies that exist in the health insurance system for the coverage of breast cancer treatment.

## Conclusion

The present study has illuminated the caregiving motivations and experiences of caregivers of women living with advanced breast cancer in Ghana. Through the narratives of family caregivers, it was identified that socio-cultural obligations and reciprocity were key motivating factors in the choice of family members and friends to accept the caregiving role. Further, family caregivers were the key agents for the management of symptoms such as a malignant wound, pain, and lymphedema in the home due to a lack of formal support services. This finding has significant implication for policy intervention. The non-existence of integrative palliative care services in the Ghanaian socio-cultural context requires the healthcare structure to work on instituting and improving integrative palliative care services. Also the use of a collaborative-based home model for the direct provision of palliative care services is paramount. This will, in turn, require home-based support programs to assist caregivers in their caring role especially in the area of symptom management, and direct governmental social intervention programs (e.g. transportation to treatment facilities and drugs for patients) to resource-limited caregiving families of women with advanced breast cancer. The National Health Insurance Scheme should be expanded to fully cover breast cancer treatment to women and their families since the current partial subsidization of breast cancer treatment is a challenge.

### Limitations and strengths of the study

Some limitations of the current study need to be addressed. First, findings from the study reported on caregivers of women with advanced-stage breast malignancy. Hence, these findings may not apply to caregivers of women living with the early-stage disease. The focus of this study was also limited to a single site. Majority of the participants in this present study were women and therefore, the degree to which findings from this study are generalized to male caregivers should be done carefully. Therefore, it is essential for studies to specifically explore the experiences family caregivers of early-stage breast cancer as well as male caregivers in breast cancer caregiving to foster a comprehensive understanding of the phenomenon.

We recruited a diverse sample of caregivers with different characteristics such as age (ranging from 25 to 73 years), gender, religion, socioeconomic status and duration of caregiving. This allowed a full exploration of the experiences of family caregivers with diverse backgrounds. Further, the key strength of this study is its rigor. For instance, the credibility of the study was determined by using member checking to validate the study findings. We are also the first to report on the caregiving motivations and experiences among family caregivers of patients living with advanced breast cancer in the socio-cultural context of Ghana.

### What is already known about the topic?

- Most women in Ghana are usually diagnosed with advanced-stage disease.

- There is a gap in research about what motivates caregivers to assume the caregiving role and their home care experiences with caregiving.

## What does this paper add?

- Family members usually support women with advanced breast cancer in the home setting.

- This study identifies socio-cultural obligation and reciprocity as reasons for accepting the caregiving role.

- The lack of home-based palliative care services in the Ghanaian health system structure makes caregivers the key agents of management of breast cancer-related symptoms.

- Caregivers provide multi-dimensional forms of support for patients diagnosed with advanced breast cancer.

## Supporting information

**S1 File. Interview guide in original language (English).**
(DOCX)

**S2 File. Participants' quotes.**
(DOCX)

**S1 Table. Demographic characteristics of participants.**
(DOCX)

## Acknowledgments

We would like to express our profound gratitude to all the family caregivers who shared their caregiving experiences with us.

## Author Contributions

**Conceptualization:** Grace Kusi, Adwoa Bemah Boamah Mensah, Kofi Boamah Mensah.

**Data curation:** Grace Kusi.

**Formal analysis:** Grace Kusi, Adwoa Bemah Boamah Mensah.

**Investigation:** Grace Kusi, Adwoa Bemah Boamah Mensah.

**Methodology:** Grace Kusi, Adwoa Bemah Boamah Mensah, Kofi Boamah Mensah.

**Project administration:** Grace Kusi, Adwoa Bemah Boamah Mensah.

**Resources:** Grace Kusi.

**Supervision:** Adwoa Bemah Boamah Mensah.

**Validation:** Adwoa Bemah Boamah Mensah.

**Writing – original draft:** Grace Kusi, Adwoa Bemah Boamah Mensah.

**Writing – review & editing:** Grace Kusi, Adwoa Bemah Boamah Mensah, Kofi Boamah Mensah, Veronica Millicent Dzomeku, Felix Apiribu, Precious Adade Duodu.

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
