## [Decision Letter · Decision Letter 0]

13 Dec 2019

PONE-D-19-26686

Caregiving motivations and experiences among family caregivers of patients living with advanced breast cancer in Ghana.

PLOS ONE

Dear Dr Mensah,

Thank you for submitting your manuscript to PLOS ONE. After careful consideration, we feel that it has merit but does not fully meet PLOS ONE’s publication criteria as it currently stands. Therefore, we invite you to submit a revised version of the manuscript that addresses the points raised during the review process.

Please respond to the comments by reviewer 2.  In your discussion I suggest it might be better to cite research supporting your findings of family support rather than the Iranian study.

We would appreciate receiving your revised manuscript by Jan 27 2020 11:59PM. To enhance the reproducibility of your results, we recommend that if applicable you deposit your laboratory protocols in protocols.io, where a protocol can be assigned its own identifier (DOI) such that it can be cited independently in the future. For instructions see: http://journals.plos.org/plosone/s/submission-guidelines#loc-laboratory-protocols

We look forward to receiving your revised manuscript.

Kind regards,

Rosemary Frey

Academic Editor

PLOS ONE

Journal Requirements:

Please ensure that your manuscript meets PLOS ONE's style requirements, including those for file naming. The PLOS ONE style templates can be found at http://www.plosone.org/attachments/PLOSOne_formatting_sample_main_body.pdf and http://www.plosone.org/attachments/PLOSOne_formatting_sample_title_authors_affiliations.pdf

Reviewers' comments:

Reviewer's Responses to Questions

**Comments to the Author**

1. Is the manuscript technically sound, and do the data support the conclusions?

Reviewer #1: Yes

Reviewer #2: Yes

2. Has the statistical analysis been performed appropriately and rigorously? 

Reviewer #1: N/A

Reviewer #2: N/A

3. Have the authors made all data underlying the findings in their manuscript fully available?

Reviewer #1: No

Reviewer #2: Yes

4. Is the manuscript presented in an intelligible fashion and written in standard English?

Reviewer #1: Yes

Reviewer #2: Yes

5. Review Comments to the Author

Reviewer #1: Thank you for the opportunity to review this well planned and executed investigation into caregiving motivations and experiences among family caregivers of patients living with advanced breast cancer.

Specific comments follow:

It is unclear in Data analysis whether the procedure you took is the Colaizzi approach – could you clarify this. Colaizzi is difficult to find on line therefore it needs more clarification and preferably a more recent reference in addition. Also, the reference needs to show it is an edited book but consider using a more up to date reference for your methodology and relate more what you did to the reference you are using.

Line 55 – Word form - Wound

Line 58 – article required before treatment – also subject verb agreement – head noun in that sentence is treatment trend (is)

Line 64 add “as well as”

Line 72 researches (Count noun?)

Line 80 Delete article an and add an article in line 116

Line 118 delete s off interests

Line 151 count vs non-count noun -information

Line 197 – add a reference to confirmability claim

Line 204 a reference for and more detail of ‘voice-centered data analysis’ -

Table S1? Where is it?

Line 336 replace these for this

Line 345 cited that?

Line 443 – delete an ‘only’

Line 475 – 78 Not sure what you are saying here. Sense?

Discussion - could do with being a bit sharper and to the point.

Line 526- I’m not sure the contrast with the Iranian participants adds anything because we aren’t told why this is different

Line 538 – sense?

Line 542 – incomplete sentence and is it an irony?

Line 560 – which home based model are you referring to? Have you explained it elsewhere in the paper?

Reviewer #2: This article makes an important contribution to family and socio-cultural factors in breast cancer caregiving in Ghana. The paper still needs a bit more proofreading, but is otherwise satisfactory in all major respects and I would recommend publication.

6. PLOS authors have the option to publish the peer review history of their article (what does this mean?). If published, this will include your full peer review and any attached files.

Reviewer #1: No

Reviewer #2: No

---

## [Author Response · Author response to Decision Letter 0]

15 Jan 2020

Reviewer #1

1. We thank the reviewer for the important suggestion raised on clarifying the Colaizzi’s approach used for data analysis. The comment has been addressed and highlighted in green font of the revised manuscript, now page 9-10, line 189-217 to provide more clarification on how the Colaizzi’s approach was used. 

2. We thank the reviewer for this important comment. We have addressed this comment in page 3, line 55. 

3. This comment has now been addressed in the revised manuscript in page 4, now line 73

4. We thank the reviewer for the observation made. We have added as well as in the revised manuscript page 4, now line 84. 

5. This comment has been addressed in page 4, now line 93 of the revised manuscript.

6. We thank the reviewer for the observation made. We have deleted the article “an” now in page 5, line 101 of revised manuscript. 

7. The article “a” have been added to page 7, now line 137 and the sentence has been reconstructed to make it more meaningful. 

8. We thank the reviewer for the observation made. We have deleted s off interests in page 7, now line 139 of the revised manuscript. 

9. This comment has been addressed in the revised manuscript in page 8, now line 172. 

10. We thank the reviewer for this comment. We have addressed this comment, and added a reference to the confirmability claim in page 11, now line 236 of the revised manuscript.

11. We thank the reviewer for this comment. This comment has been addressed and a reference and a more meaningful statement of data analysis has been added in page 11, line 242-244 of the revised manuscript. 

12. We thank the reviewer for this comment. Table S1 is in the additional file that was submitted with the manuscript and has been highlighted in page 12, now line 263 of the revised manuscript 

13. We thank the reviewer for the important observation. We have addressed this in page 17, now line 364 of the revised manuscript. 

14. This has been well elaborated in page 17, now line 371-374 of the revised manuscript. 

15. We thank the reviewer for pointing this out. We agree with the comment. Therefore we have addressed in page 20, now line 460-461 of the revised manuscript.

16. We thank the reviewer for this comment. We have reconstructed the statement into a more meaningful statement in page 21, now line 489 -491 of the revised manuscript.

17. We appreciate the reviewer’s comments. As suggested by the reviewer, we have been sharper in the ‘discussion’ section in page 22-24, now line 504-562 of the revised manuscript. 

18. We thank the reviewer for this suggestion. the Iranian example has been removed and the statement has been amended as suggested in Page 23, now line 543-546 .

19. We thank the reviewer for the important input. We have reconstructed the sentence into a complete statement to make it more meaningful in page 27, line 554-557

20. We thank the reviewer for the important input. The Home based model have been introduced and well elaborated in the ‘introduction section (page 3-4, now line 58-77)’. This has given a context to the home-based care model presented in the “conclusion section” page 25, now line 573 of the revised manuscript .

Reviewer #2

We thank the reviewer for this valuable feedback. As suggested by the reviewer, we have accordingly proofread the revised manuscript.

---

## [Decision Letter · Decision Letter 1]

12 Feb 2020

Caregiving motivations and experiences among family caregivers of patients living with advanced breast cancer in Ghana.

PONE-D-19-26686R1

Dear Dr. Mensah,

We are pleased to inform you that your manuscript has been judged scientifically suitable for publication and will be formally accepted for publication once it complies with all outstanding technical requirements.

With kind regards,

Rosemary Frey

Academic Editor

PLOS ONE

Additional Editor Comments (optional):

Reviewers' comments:

Reviewer's Responses to Questions

**Comments to the Author**

1. If the authors have adequately addressed your comments raised in a previous round of review and you feel that this manuscript is now acceptable for publication, you may indicate that here to bypass the “Comments to the Author” section, enter your conflict of interest statement in the “Confidential to Editor” section, and submit your "Accept" recommendation.

Reviewer #1: All comments have been addressed

2. Is the manuscript technically sound, and do the data support the conclusions?

Reviewer #1: Yes

3. Has the statistical analysis been performed appropriately and rigorously? 

Reviewer #1: N/A

4. Have the authors made all data underlying the findings in their manuscript fully available?

Reviewer #1: No

5. Is the manuscript presented in an intelligible fashion and written in standard English?

Reviewer #1: No

6. Review Comments to the Author

Reviewer #1: Still some errors in the text - I've highlighted a few here, but text needs careful reading mainly for small grammar errors - subject/verb agreement, word form error (women vs female), preposition use etc. Otherwise authors have addressed previous comments. Our findings correspond with earlier research works which posit that caregivers of women

527 provided support with daily activities (14, 49).Caregivers were the main providers of emotional support

544 by offering patients with words of encouragement. this further move other women cancers?

7. PLOS authors have the option to publish the peer review history of their article (what does this mean?). If published, this will include your full peer review and any attached files.

Reviewer #1: No

---

## [Editor Report · Acceptance letter]

24 Feb 2020

PONE-D-19-26686R1 

Caregiving motivations and experiences among family caregivers of patients living with advanced breast cancer in Ghana. 

Dear Dr. BOAMAH MENSAH:

I am pleased to inform you that your manuscript has been deemed suitable for publication in PLOS ONE. Congratulations! Your manuscript is now with our production department. 

With kind regards,

on behalf of

Dr. Rosemary Frey 

Academic Editor

PLOS ONE